# More Physical Activity after Concussion Is Associated with Faster Return to Play among Adolescents

**DOI:** 10.3390/ijerph18147373

**Published:** 2021-07-09

**Authors:** Corrine N. Seehusen, Julie C. Wilson, Gregory A. Walker, Sarah E. Reinking, David R. Howell

**Affiliations:** 1Sports Medicine Center, Children’s Hospital Colorado, Aurora, CO 80045, USA; corrine.seehusen@childrenscolorado.org (C.N.S.); julie.wilson@childrenscolorado.org (J.C.W.); gregory.walker@childrenscolorado.org (G.A.W.); 2Department of Orthopedics, University of Colorado School of Medicine, Aurora, CO 80045, USA; 3Department of Pediatrics, University of Colorado School of Medicine, Aurora, CO 80045, USA; 4Physical Therapy Department, Children’s Hospital Colorado, Aurora, CO 80045, USA; sarah.reinking@childrenscolorado.org

**Keywords:** mild traumatic brain injury, pediatric, rehabilitation, exercise

## Abstract

Concussion management has evolved to de-emphasize rest in favor of early introduction of post-injury physical activity. However, the optimal quantity, frequency and intensity of physical activity are unclear. Our objective was to examine the association between objective physical activity metrics and clinical recovery after concussion. We prospectively enrolled a cohort of 32 youth athletes with concussion, evaluated within 14 days of injury and after return-to-play (RTP) clearance. For two weeks after enrollment, we quantified steps/day and exercise frequency, duration, and intensity via wrist-worn actigraphy. We grouped participants by RTP clearance timing (<28 days vs. ≥28 days). Seventeen (53%) participants required ≥28 days post-concussion for RTP clearance. Groups were similar in age (14.9 ± 1.9 vs. 15.4 ± 1.5 years; *p* = 0.38), proportion of females (47% vs. 40%; *p* = 0.69), and concussion history (59% vs. 47%; *p* = 0.49). During the study period, the RTP ≥ 28 days group took fewer steps/day (8158 ± 651 vs. 11,147 ± 4013; *p* = 0.02), exercised fewer days/week (2.9 ± 2.4 vs. 5.0 ± 1.9 days/week; *p* = 0.01), and exercised fewer total minutes/week (117 ± 122 vs. 261 ± 219 min/week; *p* = 0.03). Furthermore, we observed ≥10,251 average steps/day, ≥4 exercise sessions/week, and exercising ≥134 total minutes/week were optimal cutpoints to distinguish between groups. These findings support the benefit of physical activity during concussion recovery.

## 1. Introduction

Concussions are a subset of mild-traumatic brain injury (mTBI) that result in a disruption of neuronal functions due to trauma [1], and result in a wide array of symptoms (e.g., headache, fatigue, or irritability [2]). Concussions are common in sports among all age groups from youth to professional athletes [3,4]. While many adolescent athletes recover within a month [5], some have persistent symptoms, resulting in extended time away from sport, school, or social activities [6]. Beyond the risk of persistent symptoms, those who have sustained a concussion have an increased risk of sport-related musculoskeletal injury. One study reported that adolescent athletes returning to sport after their concussion had a 34% increased risk of subsequent musculoskeletal injury than those who did not sustain a concussion, even after they have received return to play clearance (i.e., permission to return to full contact game play or participation) from a medical provider [7]. This increased musculoskeletal injury risk may be linked in part to physical deconditioning as a result of removal from sport [8], and as such, physical activity during concussion recovery may be one approach to help mitigate subsequent injury risk.

Previously, absolute rest until complete symptom resolution was recommended as the standard of care for patients post-concussion [2,9], although this was originally described as management for patients who recovered within 7 to 10 days [10]. However, more recent studies have reported that strict rest outside of the acute window of 24 to 48 h post-injury may worsen outcomes or prolong recovery time [11,12]. Furthermore, additional research has demonstrated that aerobic exercise below the level of symptom exacerbation may promote faster recovery [13]. As a result, management guidelines have evolved to not only de-emphasize rest but also promote early introduction of physical activity and cognitive activity that does not worsen symptoms in early recovery [2]. Current recommendations provided by the most recent consensus statement on concussion in sport suggest that after a brief (24–48 h) period of strict rest post-injury, clinicians should recommend gradually adding more activity that does not exacerbate symptoms in a graduated and stepwise fashion [2]. Upon symptom resolution, patients are then recommended to progressively increase exercise demands through a stepwise strategy as long as concussion-related symptoms do not return. Although sub-symptom threshold aerobic activity during recovery is linked to faster recovery times [13,14,15], the optimal quantity, frequency and intensity of physical activity levels during concussion recovery have yet to be determined and formally outlined.

The purpose of this study was to investigate the association of return to play (RTP) timing with objectively recorded physical activity and exercise metrics during concussion recovery. We hypothesized more physical activity, more intense exercise sessions, and higher volumes of exercise would be associated with shorter RTP clearance times.

## 2. Materials and Methods

### 2.1. Study Design

We conducted an observational, prospective cohort study of youth athletes who sustained a concussion and were initially evaluated by Children’s Hospital Colorado board certified sports medicine physicians between 11 April 2019 and 27 October 2020. Prior to study commencement, the Colorado Multiple Institutional Review Board approved the protocol, and informed consent/assent was provided by participants and their parents (if <18 years of age) before enrollment in the study. Participants were evaluated at two time points: at their initial visit (<14 days post-injury) and at their return to play (RTP) clearance visit. Diagnoses and RTP decisions were made by sports medicine physicians, consistent with the criteria outlined in the most recent consensus statement on concussion in sport during the study period [2], independent of any study procedures.

Participants reported concussion symptom severity at each visit using the Post-Concussion Symptom Inventory (PCSI). Between visits, each participant wore an activity tracking device (Fitbit Charge 3™, Fitbit Inc, San Francisco, CA, USA) that monitored a variety of physical activity and exercise parameters. For the purposes of this analysis, we calculated the average number of steps per day, average number of exercise sessions per week determined by number of entries manually or automatically registered by the device (exercise frequency), average time spent during each specific exercise session (exercise duration), and average and maximum HR during each session (exercise intensity). We elected to analyze data that were collected over the first two weeks after the initial visit to provide consistent time-specific information post-injury that aligned with our primary end point (28 days after concussion).

### 2.2. Participants

We recruited participants who were seen and cared for at a single sports medicine center. Inclusion criteria included being diagnosed with a concussion, presenting for care within 14 days of injury, being 12–18 years of age, being diagnosed with a concussion by a board-certified sports medicine physician, having a PCSI score ≥ 9 during the first assessment to ensure participants had not recovered by the time they enrolled in the study, and planning to continue in organized sport participation after RTP clearance. We excluded participants if they had a concurrent lower extremity injury that would affect balance, had a previous concussion within the last six months, a second head impact before clearance from the initial concussion, a pre-existing learning disability, documented structural brain injury via neuroimaging, or sustained a concussion during a high velocity mechanism (e.g., motor vehicle accident).

### 2.3. Patient-Reported Outcomes

To assess concussion symptoms, participants completed the Post-Concussion Symptom Inventory (PCSI) at their initial and RTP assessments. The PSCI has been developed and validated specifically within the youth and adolescent population [16,17], and contains strong internal consistency (α = 0.94) and test-retest reliability (ICC = 0.79). Participants responded to 21 items by rating each symptom (e.g., headache, nausea, balance problems) on a 7-point scale indicating severity from 0 (Not a Problem) to 6 (Severe Problem). We calculated total PCSI score as the sum of all responses where the higher the value, the more symptomatic the participant [18,19].

### 2.4. Physical Activity and Exercise Data

After the initial assessment, participants received a Fitbit Charge 3™ and were instructed to wear the device at all times to track physical activity from the first full day after the initial assessment until they received RTP clearance. Each participant was given a pre-selected account login and password specific for the purposes of this study. The research staff created each individual account so that both researchers and participants could log in and review the information from the device phone and web application without the participant needing to provide personal information. A member of the research staff would log into the account of each participant to verify proper setup and recording. If the patient was clearly not wearing the device, noted if there were long periods during the day without steps recorded or whole days with less than 300 steps, the research staff member would contact the family by phone or email to remind them to wear the device and sync it through the phone application daily. On a weekly basis during the study period, members of the research staff would access the account of each participant and extract exercise and physical activity parameters including number of steps per day, number of weekly exercise sessions, exercise mode, duration of each session in minutes, maximum heart rate (HR) during each session, and the average HR for each session. If there was a day where the participant was not compliant with the instructions provided, that day was removed from the analysis and the values were calculated based on the data provided.

To determine step count, the device uses a 3-axis accelerometer to count steps relying on arm movement during normal gait [20]. The exercise session type was either determined manually on the device by the participant before beginning exercise or the device would automatically recognize and record high-movement activities through the SmartTrack™ feature. Participants could manually select from six different modes: run, bike, treadmill, swim, weights, and walk. If not manually started, the SmartTrack™ feature would recognize activities of at least 15 min in length and categorize it as a walk, run, sport, or aerobic workout [21]. A systematic review on wearable devices determined that this measurement approach demonstrated acceptable validity (±10% measurement error) and reliability (correlation coefficient = 0.75) [22].

### 2.5. Statistical Analysis

Data are presented as means (standard deviations) or medians for continuous outcome variables, and as the number included (corresponding percentage) within each group for categorical outcome variables. We compared groups (RTP < 28 days vs. ≥28 days) on physical activity (average steps per day), exercise frequency (average workouts per week), exercise duration (average time for each workout), exercise volume (average total number of minutes per week), and intensity (average HR during exercise, and average peak HR during workouts) during the first two weeks after the initial visit using independent samples *t*-tests, and calculated effect sizes using Cohen’s d values. A time period of two weeks was selected as the participants were most compliant in wearing the device in the first two weeks after the initial assessment. Additionally, this time period is consistent with another study in a similar population [23]. We then used a Receiver Operating Characteristic (ROC) and Area under the Curve (AUC) analysis to identify different cutpoint thresholds to distinguish between groups for each exercise/activity variable. Using a stepwise approach, we identified the sensitivity and specificity for the value, which demonstrated the highest classification accuracy to distinguish between groups for each variable. Missing data were treated as such, and no imputations were performed. Before the analysis during the manual data extraction, the exercise sessions and step count points were screened for validity by a member of the research team. Data that was deemed as incomplete (e.g., a started exercise session that lasted only a few seconds or steps recorded for only a one-hour period in totality for the day) was treated as missing. All statistical tests were two-sided and performed using Stata Statistical Software: Version 15 (StataCorp, LLC, College Station, TX, USA).

## 3. Results

We enrolled a total of 32 participants who completed their initial assessment <14 days post-concussion and underwent activity monitoring until they were cleared by their physician to RTP. Within the cohort, 17 (53%) required ≥28 days to receive RTP clearance. The two groups were of similar ages, proportion of females, and past concussion history (Table 1). Those who required ≥28 days for RTP clearance were enrolled at a similar time post-injury but reported significantly greater symptom severity at the initial assessment than the group who received RTP clearance within 28 days of injury (Table 1). At the RTP clearance visit, both groups reported similar symptoms scores on the PCSI (Table 1). In addition, those in the RTP ≥ 28 days post-injury group reported more severe initial symptoms across physical (29.1 ± 8.0 vs. 18.0 ± 11.6, *p* = 0.004), cognitive (19.1 ± 6.8 vs. 12.3 ± 8.9, *p* = 0.02), emotional (10.4 ± 6.2 vs. 4.9 ± 4.6, *p* = 0.009), and fatigue (9.9 ± 4.7 vs. 5.5 ± 4.3, *p* = 0.01) domains.

During the first two weeks after study enrollment, the group who received RTP clearance ≥ 28 days post-injury took significantly fewer steps per day (Figure 1A), exercised significantly fewer days per week (Figure 1B), and had significantly lower average total exercise volume per week (Figure 1D) compared with the group who received RTP clearance < 28 days post-injury. No significant between-group differences were observed for average amount of time spent exercising per session (Figure 1C). There were no significant differences in exercise intensity (average HR during exercise or peak HR during exercise) between groups (Figure 2).

When examining the ability of physical activity and each exercise variable to distinguish between RTP < 28 days vs. ≥28 days groups, the highest classification accuracy between groups was ≥10,250 average steps/day (Table 2). In addition, completing ≥4 exercise sessions per week and ≥135 total minutes per week exercising demonstrated moderate classification accuracy (>70%; Table 2).

## 4. Discussion

Data from our cohort study indicate that objectively measured physical activity and exercise metrics are associated with recovery time following concussion among pediatric and adolescent athletes. Specifically, those who recovered within 28 days of injury engaged in more exercise or physical activity in the two weeks after the initial evaluation. However, due to the nature of our study design, we cannot determine casualty based on our findings.

Although those who required ≥28 days to receive RTP clearance were tested at a similar timepoint initially, they reported significantly greater symptom severity at the initial assessment than the group who received RTP clearance within 28 days of injury, which may have affected subsequent recovery trajectories regardless of the level of physical activity they performed. This aligns with previous research that consistently describes higher initial concussion symptom rating, whether on the first day or initial few days after injury, is a predictor of prolonged concussion recovery [2,6,24]. Furthermore, while it is plausible that more exercise after concussion contributed to a faster recovery, it is also just as likely that those who were feeling better early on after injury were more likely to engage in more consistent exercise and physical activity. Despite this, our findings build on past work that used self-reported exercise volume suggesting further that aerobic exercise is associated with faster recovery after concussion [14,18].

During the first two weeks following the initial evaluation, the group who went on to receive RTP clearance ≥28 days post-injury took significantly fewer steps per day and exercised significantly fewer days per week compared with the group who received RTP clearance <28 days post-injury. The median step count per day over the two-week period was 10,893 for those who received clearance <28 days and 7571 for those who took ≥28 days to receive RTP clearance. Interestingly, the values obtained for those who took ≥28 days to receive RTP clearance are similar to a past study, where researchers observed an average count of approximately 6600 steps/day in the days following a concussion [23]. Furthermore, a systematic review reported normative values among boys and girls to range on average 12,000 to 16,000 steps/day and 10,000 to 13,000 steps/day, respectively [25]. Across our sample, the majority of adolescents with a concussion were typically below this normative level (67% of boys, 64% of girls). In a study that compared concussed to non-concussed high school and collegiate football players, researchers observed that there was a significant difference in step count for the first 2 days post-concussion between players who were concussed to those who were not. Specifically, the authors reported that those who had a concussion averaged approximately 6663 steps per day compared to the controls who averaged approximately 11,148 steps per day [23]. In our cohort, those who received clearance in <28 days had an average step count per day closer in value to these non-concussed athletes, while those who took ≥28 days to receive RTP clearance were more similar in daily step count value to the acutely concussed football players. We did not examine when these steps were taken throughout the day or divide the participants based on sport participation. If following the recommended RTP guidelines recommended by their medical provider, many should have resumed physical activity in a step-wise manner by participating in conditioning or a non-contact practice. Depending on each participant’s specific sport, the number of steps per day could be highly impacted by one, high-set volume sport session (i.e., going on a run or attending non-contact soccer practice).

The group who received RTP clearance ≥28 days post-injury also exercised for fewer total minutes per week than those who recovered within the first 28 days of concussion. As with our step count findings, these are interesting association-level data that suggest a link between exercise volume and concussion recovery, aligning with past work [18]. However, we did not observe significant between-group differences for the average amount of time spent exercising or exercise intensity per session, suggesting that exercise frequency may be an important element of exercise prescription during concussion recovery.

At the RTP clearance visit, both groups reported similar symptoms severity; however, those who took <28 days for RTP clearance had a lower but non-significant mean PCSI rating at the RTP visit. Although not statistically significant, the difference between the averages may show some clinical significance (mean difference = 10 points, 95% CI = −3, 23). As with prior work, this indicates that many patients recovering from a concussion may still have “concussion-like” symptoms when receiving RTP clearance, which reflects the non-specific nature of symptom inventories [17]. Additionally, current symptom severity at the RTP clearance assessment for both the <28 days and ≥28 days clearance groups, PSCI = 12.3 (13.8) and 2.5 (6.6) respectively, were greater than 0, meaning that RTP clearance is not synonymous with overall symptom resolution. This non-significant, but potentially clinically significant finding may help explain why the ≥28 group took longer to receive RTP clearance. The standard-of-care gradual RTP protocol recommends that the athlete should progress to the next step of more intensive exercise only if they are asymptomatic at the current stage. If the symptoms are not resolving, they may not continue on to the next stage, possibly suggesting that the symptoms may be driving more sedentary behavior out of compliance with recommendations.

When observing the recorded HR during each exercise session, there were no significant group differences in average HR or peak HR during exercise. When considering exercise prescriptions in the context of post-concussion management, clinicians should consider the intensity, frequency, and duration [26]. Our collective approach indicates that the volume of exercise was more strongly associated with prolonged recovery than the intensity of said exercise. This may be due to several reasons. Namely, the actigraphy device we used, while beneficial for step counting, may be limited in its ability to quantify heart rate, particularly at higher intensity levels [22], mitigating any potential dose-response effects we may have observed. In addition, as this was an observational study, we did not dictate the intensity level for participants. Instead, consistent current recommendations for concussion management [2], these participants were told to perform symptom-limited physical activity. One recent randomized controlled trial indicated that a prescription of aerobic exercise led to a faster symptom resolution time compared to a placebo stretching protocol [14]. However, intensity and volume information were not provided. Therefore, while exercise may be beneficial, further work needs to examine the optimal physical activity or exercise dose to facilitate optimal healing.

When examining the ability of physical activity and each exercise variable to distinguish between RTP <28 days vs. ≥28 days groups, the highest classification accuracy between groups was ≥10,250 average steps/day. In addition, completing ≥4 average exercise sessions per week and ≥135 total minutes per week exercising demonstrated classification accuracy >70%. Daily step count is highly influenced by periods of activity (i.e., going on a walk would add to one’s step count significantly). Given that following concussion, patient physical activity is symptom-limited, step counts may be a reasonable metric to examine further to ensure individuals do not engage in “cocoon therapy” [27]. However, further work is required to understand if a low-level activity prescription (i.e., a step count strategy) results in faster recovery compared to a higher intensity exercise recommendation (i.e., exercise prescription based on HR). One study compared adults who were randomized to a prescription of 30 min of daily light exercise vs. gradual return to exercise after symptom resolution and found no differences in recovery outcomes [28]. Thus, low level intensity/volume exercise may also not provide the necessary dose to induce a meaningful clinical benefit especially in a population of highly active adolescents. Nonetheless, our data may be a useful guide for clinicians involved in concussion management to help set activity goals for their patients.

### Limitations

Although we found that those who received RTP clearance within 28 days had more active minutes per week and took more steps per day than those who took longer to receive RTP clearance, our study was not designed to determine causality. Through our observational study we cannot determine if those who were more active received clearance sooner as a result of their physical activity and exercise levels or if there were other factors influencing their RTP clearance time. Participants with faster RTP clearance engaged in more physical activity and exercise during recovery, but also had a lower concussion symptom burden initially post-injury, so they may have been able to tolerate more physical activity and exercise sooner. In addition, although instructed to wear the activity tracking device at all times, we were unable to fully ensure compliance. While research staff could monitor incomplete data and urge participants to continue wearing it through the two-week period, we cannot be entirely sure that all steps or active minutes were registered. Additionally, some of the sessions were recorded via the device SmartTrack™ function and were not manually initiated. As a result, we cannot ensure that every mode was accurately determined or the entire validity of every session. Our study was conducted at a single center specialty clinic, which may create geographic and referral biases in addition to contributing to the lack of racial and ethnic diversity in our cohort. Individual variability within physician clinical decision-making related to RTP clearance may have also affected our findings. Finally, given the relatively small sample size, the low sensitivity and specificity we observed may have been due several factors, including the low power of measurement in the instruments used.

## 5. Conclusions

Among adolescents with concussion, we observed an association between returning to unrestricted sports participation within 28 days of injury with more steps/day, exercise sessions/week, and spending more total time exercising/week. These preliminary results from our cohort study further support the potential benefits of physical activity during concussion recovery, although our study is designed to evaluate association, not causation. Further research is needed to prospectively examine the duration-, frequency- and intensity-specific physical activity levels found in this study to aid clinicians in concussion management.

## Figures and Tables

**Figure 1 ijerph-18-07373-f001:**
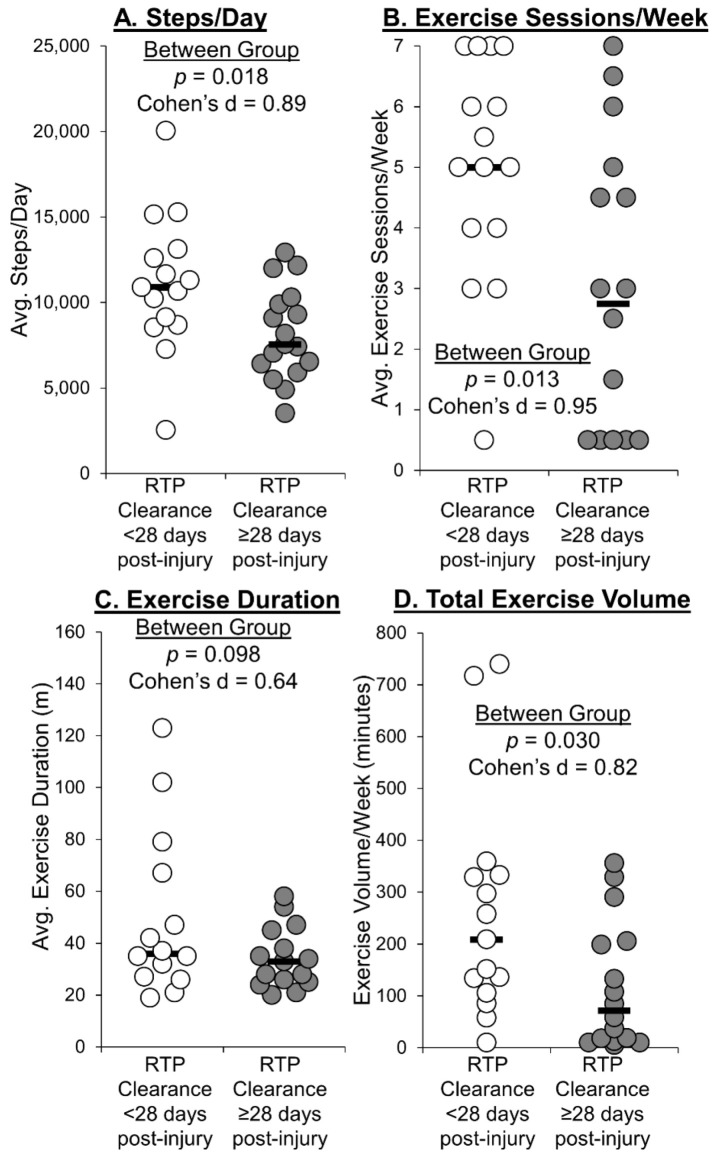
Individual data points describing the exercise characteristics (**A**) average steps per day; (**B**) average exercise sessions per week; (**C**) average duration when exercising; (**D**) average total exercise volume per week. across the two weeks after study enrollment between those who received RTP clearance 28 days or greater post-injury compared to those who received RTP clearance in less than 28 days post-injury. Black bars represent the group mean.

**Figure 2 ijerph-18-07373-f002:**
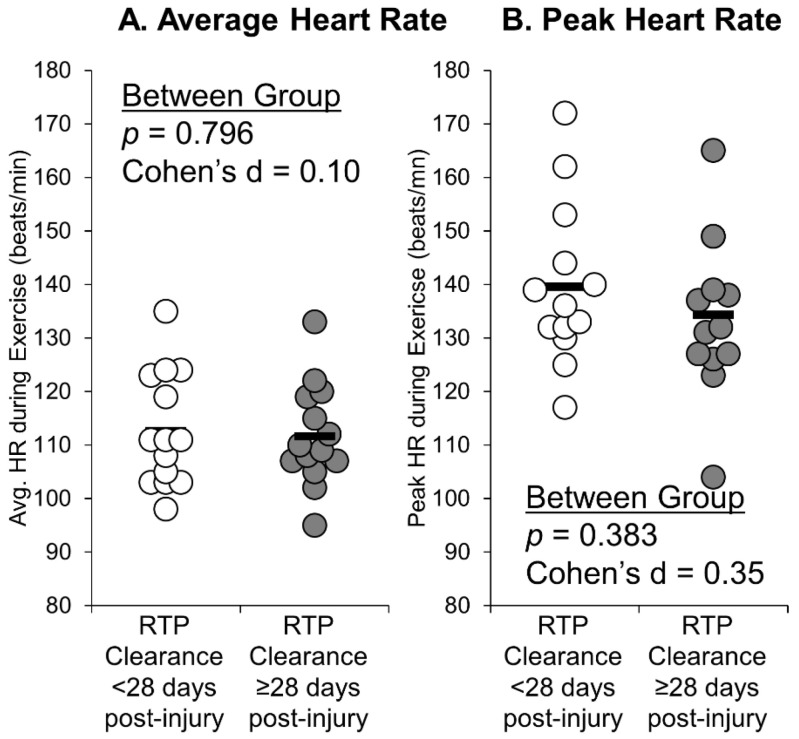
Individual data points describing the average (**A**) and peak (**B**) heart rate during exercise during the two weeks after study enrollment between those who received RTP clearance 28 days or greater post-injury compared to those who received RTP clearance in less than 28 days post-injury. Black bars represent the group mean.

**Table 1 ijerph-18-07373-t001:** Demographic and clinical characteristics between groups. Represented as means (standard deviation).

Variable	RTP ≥ 28 Days Post-Injury (*n* = 17)	RTP < 28 Days Post-Injury (*n* = 15)	*p*-Value
Age (years)	14.9 (1.9)	15.4 (1.5)	0.38
Sex (female)	8 (47%)	6 (40%)	0.69
Race (white)	13 (76%)	10 (67%)	0.70
History of concussion	10 (59%)	7 (47%)	0.49
Number of prior concussions	One: 4 (24%) Two: 6 (35%)	One: 4 (27%) Two: 2 (13%) Five: 1 (7%)	-
Height (cm)	164.4 (7.8)	168.4 (10.9)	0.23
Mass (kg)	55.7 (10.0)	66.1 (15.6)	0.03
Initial assessment timing (days post-injury)	8.2 (3.1)	6.5 (2.4)	0.09
RTP clearance assessment timing (days post-injury)	71.7 (33.9)	20.1 (10.5)	-
Symptom severity at initial assessment (PCSI score)	70.2 (22.3)	42.9 (24.9)	0.003
Symptom severity at RTP clearance assessment (PCSI score)	12.3 (13.8)	2.5 (6.6)	0.15

**Table 2 ijerph-18-07373-t002:** Receiver operating characteristic, sensitivity, specificity, and classification accuracy to determine whether participants were cleared to RTP within 28 days after concussion for each exercise outcome variable.

Variable	AUC Value (95% Confidence Interval)	Sensitivity	Specificity	Classification Accuracy
≥10,250 average steps per day	0.75 (0.57, 0.93)	66.7%	76.5%	71.9%
≥4 average exercise sessions per week	0.74 (0.56, 0.93)	80.0%	62.5%	71.0%
≥35 min per exercise session	0.62 (0.41, 0.84)	64.3%	60.0%	62.1%
≥135 total min per week exercising	0.73 (0.55, 0.92)	73.3%	68.8%	71.0%
≥132 beats/min peak HR during exercise	0.60 (0.38, 0.83)	76.9%	46.2%	61.5%
≥111 beats/min average HR during exercise	0.52 (0.30, 0.75)	57.1%	57.1%	57.1%

## Data Availability

Deidentified individual patient data and other study-related documents can be shared upon request via the corresponding author.

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
