# Peer review of "More Physical Activity after Concussion Is Associated with Faster Return to Play among Adolescents"

_ijerph, 2021, doi:10.3390/ijerph18147373_

Round 1

Reviewer 1 Report

This is a good paper, on a valuable topic.  I wish the authors had included some information on the type of symptoms reported by the >28 RTP group. It is possible that there was a pattern of symptoms (e.g., symptoms suggestive of vestibular dysfunction) that could account for the increased length of recovery.  If there was no review of the symptoms between the two groups, this should be listed as a weakness.  

Author Response

This is a good paper, on a valuable topic.  I wish the authors had included some information on the type of symptoms reported by the >28 RTP group. It is possible that there was a pattern of symptoms (e.g., symptoms suggestive of vestibular dysfunction) that could account for the increased length of recovery.  If there was no review of the symptoms between the two groups, this should be listed as a weakness. 

Author response: We agree with the reviewer that more detail on the types of symptoms by the group who experienced a recovery duration >28 days would be useful. Thus, we have added initial symptom domain data to our results section, based on the original symptom domains derived by Sady and colleagues (2014) on the Post-Concussion Symptom Inventory.

Lines 168-172

In addition, those in the RTP ≥28 days post-injury group reported more severe initial symptoms within physical (29.1±8.0 vs. 18.0±11.6, p=0.004), cognitive (19.1±6.8 vs. 12.3±8.9, p=0.02), emotional (10.4±6.2 vs. 4.9±4.6, p=0.009), and fatigue (9.9±4.7 vs. 5.5±4.3, p=0.01) domains.

Reviewer 2 Report

I believe that the topic of the study should be of interest to the readers in International Journal of Environmental Research and Public Health. Contents of each chapter, introduction, method, result, and discussion are very constructive. Research methods are feasible and the results are convincing. The present manuscript is an applied article with regard to physical activity after concussion applicability. Several mistakes or unclear parts have been identified. The authors must interpret their findings in the context of supporting/not supporting existing literature and provide adolescents after concussion applicability. However, I believe the authors need to revise some of the contents of the paper.

  1. Page 1, lines 25.

Abstract

The keywords should be lowercase.

  1. Page 2, lines 50.

Introduction

The author mentions “Current recommendations provided by the most recent consensus statement on concussion in sport suggest that after a brief (24-48 hour) period of strict rest post-injury”. Please specify the reference and explain which activities are suitable for post-injury with a concussion.

  1. Page 2, lines 63.

Method

In the method chapter, the present manuscript provides very constructive content of subject recruitment. The study shows that all experiments were following standard protocol. Comparing with papers in other journals, this article is an exceptionally outstanding one in method regard.

  1. Page 2, lines 77.

Method

About tracking device (Fitbit Charge 3™). How to confirm the time a child wears each day?

  1. Page 4, lines 160.

Reuslt

The P-Value should be lowercase.

  1. Page 4, lines 160.

Reuslt

The presentation of research results is rigorous and clear.

  1. Page 4, lines 160.

Discussion

Although the present manuscript provided very convincing data in the result section, the author did not elaborate his or her own study findings deeply. Most of the discussions were referred to previous studies, which have a weak connection with his or her own study findings. More elaboration of major findings conducted by this study is highly recommended.

Author Response

I believe that the topic of the study should be of interest to the readers in International Journal of Environmental Research and Public Health. Contents of each chapter, introduction, method, result, and discussion are very constructive. Research methods are feasible and the results are convincing. The present manuscript is an applied article with regard to physical activity after concussion applicability. Several mistakes or unclear parts have been identified. The authors must interpret their findings in the context of supporting/not supporting existing literature and provide adolescents after concussion applicability. However, I believe the authors need to revise some of the contents of the paper.

Author response: Thank you for your comments. We have expanded our discussion to better interpret our findings in the context of existing literature and clinical applicability in the discussion.

Lines 221-229

The median step count per day over the two-week period was 10,893 for those who received clearance <28 days and 7,571 for those who took ≥28 days to receive RTP clearance. Interestingly, the values obtained for those who took ≥28 days to receive RTP clearance are similar to a past study, where researchers observed an average count of approximately 6,600 steps/day in the days following a concussion [29]. Furthermore, a systematic review reported normative values among boys and girls to range on average 12,000 to 16,000 steps/day and 10,000 to 13,000 steps/day respectively[25]. Across our sample, the majority of adolescents with a concussion were typically below this normative level (67% of boys, 64% of girls).

Lines 261-267

This non-significant, but potentially clinically significant finding may help explain why the ≥28 group took longer to receive RTP clearance. The standard-of-care gradual RTP protocol recommends that the athlete should progress to the next step of more intensive exercise only if they are asymptomatic at the current stage. If the symptoms are not re-solving, they may not continue on to the next stage possibly suggesting that the symptoms may be driving more sedentary behavior out of compliance with recommendations.  

Page 1, lines 25.

Abstract

The keywords should be lowercase.

Author response: Thank you for pointing this out. We have corrected this in the revised manuscript (line 25).

Page 2, lines 50.

Introduction

The author mentions “Current recommendations provided by the most recent consensus statement on concussion in sport suggest that after a brief (24-48 hour) period of strict rest post-injury”. Please specify the reference and explain which activities are suitable for post-injury with a concussion.

Author response: Thank you for pointing out our omission of a reference. We have added this to the revised manuscript, as well as further explanation regarding how activity progression is recommended by expert consensus at this time.

Lines 50-54

Current recommendations provided by the most recent consensus statement on concussion in sport suggest that after a brief (24-48 hour) period of strict rest post-injury, clinicians should recommend gradually adding more activity that does not exacerbate symptoms in a graduated and stepwise fashion [2].

Page 2, lines 63.

Method

In the method chapter, the present manuscript provides very constructive content of subject recruitment. The study shows that all experiments were following standard protocol. Comparing with papers in other journals, this article is an exceptionally outstanding one in method regard.

Author response: Thank you for this comment. We are pleased to hear this, as our goal was to be transparent and clear regarding our methodology.

Page 2, lines 77.

Method

About tracking device (Fitbit Charge 3™). How to confirm the time a child wears each day?

Author response: In order to confirm the amount of time the child wore the device each day, we remotely viewed their data on a daily basis during the study monitoring period. If non-compliance was identified, we reached out to the family to ensure they had not encountered technical issues. We have expanded our explanation of this process in our revised methods section.

Lines 115-120:

A member of the research staff would log into the account of each participant to verify proper setup and recording. If the patient was clearly not wearing the device, noted if there were long periods during the day without steps recorded or whole days with less than 300 steps, the research staff member would contact the family by phone or email to remind them to wear the device and sync it through the phone application daily.

Page 4, lines 160.

Result

The P-Value should be lowercase.

Author response: Thank you for pointing this out. We have corrected as suggested.

Page 4, lines 160.

Result

The presentation of research results is rigorous and clear.

Author response: Thank you for this comment.

Page 4, lines 160.

Discussion

Although the present manuscript provided very convincing data in the result section, the author did not elaborate his or her own study findings deeply. Most of the discussions were referred to previous studies, which have a weak connection with his or her own study findings. More elaboration of major findings conducted by this study is highly recommended.

Author response: Thank you for this comment. We have elaborated in the discussion to better connect our findings to prior studies as well as to clinical practice.

Lines 223-229

Interestingly, the values obtained for those who took ≥28 days to receive RTP clearance are similar to a past study, where researchers observed an average count of approximately 6,600 steps/day in the days following a concussion [29]. Furthermore, a systematic review reported normative values among boys and girls to range on average 12,000 to 16,000 steps/day and 10,000 to 13,000 steps/day respectively[25]. Across our sample, the majority of adolescents with a concussion were typically below this normative level (67% of boys, 64% of girls).

Lines 237-244

We did not examine at when these steps were taken throughout the day or divide the participants based on sport participation. If following the recommended RTP guidelines recommended by their medical provider, many should have resumed physical activity in a step-wise manner by participating in conditioning or a non-contact practice. Depending on each participant’s specific sport, the number of steps per day could be highly impacted by one, high-set volume sport session (i.e. going on a run or attending non-contact soccer practice).

Lines 261-267

This non-significant, but potentially clinically significant finding may help explain why the ≥28 group took longer to receive RTP clearance. The standard-of-care gradual RTP protocol recommends that the athlete should progress to the next step of more intensive exercise only if they are asymptomatic at the current stage. If the symptoms are not re-solving, they may not continue on to the next stage possibly suggesting that the symptoms may be driving more sedentary behavior out of compliance with recommendations.

Reviewer 3 Report

This is an interesting study and demonstrates again that sports involves a risk of concussion, which is not enough discussed and studied by sports organisations.

lines 100-102 The PCSI is your key instrument. Please provide detail from the literature of the reliability and validity.

Another key instrument you used is RTP which you describe as a consensus assessment form administered by Board-certified sports physicians. Please provide the detailed RTP form and the assessments. What was the cut off  for RTP?

Another key variable is the instructions to each athlete after their concussion. What were they?

Another key variable is the sport (or sports) the athlete practices. How many athletes had sports which normally involved many steps in training or few and it would be expected that they would resume their previous step pattern (though diminished).

Another key variable is the history of concussions. In the group of 17 athletes with RTP > 28 days 10 (59%) had a concussion history and in the 15 with RTP < 28 days 7 (47%).  How many had 1, 2 3, 4... prior concussions and what was the RTP after each one?

Your study has tiny numbers and is more like a research report of methods (and you have described the methods you did use well. Please emphasise the small numbers and consequently very guarded conclusions in the abstract, text and conclusions and the fact that this is a cohort study. 

Author Response

This is an interesting study and demonstrates again that sports involves a risk of concussion, which is not enough discussed and studied by sports organisations.

lines 100-102 The PCSI is your key instrument. Please provide detail from the literature of the reliability and validity.

Author response: Thank you for pointing this out. We have added the internal consistency and test-retest reliability values from existing work.

Lines 101-103

The PSCI has been developed and validated specifically within the youth and adolescent population [16,17], and contains strong internal consistency (α=0.94) and test-retest reliability (ICC=0.79).

Another key instrument you used is RTP which you describe as a consensus assessment form administered by Board-certified sports physicians. Please provide the detailed RTP form and the assessments. What was the cut off for RTP? Another key variable is the instructions to each athlete after their concussion. What were they?

Author response: Participants in the study were each treated by a group of board-certified sports medicine physicians who followed the same protocol, which is based upon the guidance provided by the 5th International Consensus Statement on Concussion in Sport. Given this statement’s language, there is no specific score or cut off for RTP decision-making. Instead, it is a clinical decision made by the individual physician. Instructions provided to each athlete after their concussion were standardized as well, through a set of printed instructions provided to the patient and their family following the initial visit. We feel confident in the homogeneity of our physician practice patterns, given the routine interaction among our group. In addition, when examining the time from symptom resolution to RTP clearance, our physicians each demonstrated similar times. However, we recognize individual physician variability may have contributed to our findings, and as such, have added this as a limitation.

Lines 318-320

Individual variability within physician clinical decision-making related to RTP clearance may have also affected our findings.

Another key variable is the sport (or sports) the athlete practices. How many athletes had sports which normally involved many steps in training or few and it would be expected that they would resume their previous step pattern (though diminished).

Author response: Thank you for raising this point. We agree that this is an interesting concept to frame our findings. While our study was not designed to address the expected amount of steps in training for our participants, we have added more discussion around this point.

Lines 237-244

We did not examine at when these steps were taken throughout the day or divide the participants based on sport participation. If following the recommended RTP guidelines recommended by their medical provider, many should have resumed physical activity in a step-wise manner by participating in conditioning or a non-contact practice. Depending on each participant’s specific sport, the number of steps per day could be highly im-pacted by one, high-set volume sport session (i.e. going on a run or attending non-contact soccer practice).

Another key variable is the history of concussions. In the group of 17 athletes with RTP > 28 days 10 (59%) had a concussion history and in the 15 with RTP < 28 days 7 (47%).  How many had 1, 2 3, 4... prior concussions and what was the RTP after each one?

Author response: We agree, the number of prior concussions in each group is a useful variable to include in addition to the yes/no concussion history classification. We have modified Table 1 to now include this information.

Your study has tiny numbers and is more like a research report of methods (and you have described the methods you did use well. Please emphasise the small numbers and consequently very guarded conclusions in the abstract, text and conclusions and the fact that this is a cohort study.

Author response: Thank you for your comments. We have added the small sample size as a limitation. In addition, we have revised our abstract, first discussion paragraph, and conclusion to emphasize that this is a cohort study.

Lines 13-14

We prospectively enrolled a cohort of 32 youth athletes with concussion, evaluated within 14 days of injury and after return-to-play (RTP) clearance.

Lines 200-202

Data from our cohort study investigation indicates that objectively measured physical activity and exercise metrics are associated with recovery time following concussion among pediatric and adolescent athletes.

Lines 320-322

Finally, given the relatively small sample size, the low sensitivity and specificity we observed may have been due several factors, including the low power of measurement in the instruments used.

Lines 326-328

These preliminary results from our cohort study further support the potential benefits of physical activity during concussion recovery, although our study is designed to evaluate association, not causation.

Reviewer 4 Report

The manuscript is devoted to a topic that is beneficial for practice. The manuscript is processed well. I consider the only "shortcoming" to be the excessive brevity of all passages of the manuscript and the relatively low number of probands. As for other shortcomings or limits of work, the authors are fully aware of them and state them in passage 4.1

My only recommendation is - to expand the introduction with more emphasis on the state of the topic, the work would benefit from the inclusion of higher number of sources.

The second recommendation concerns figures - their quality does not seem to be fully satisfactory, but it is possible that in the case of a final publication, better quality will be used.

After a precise reading of the entire manuscript and making minor adjustments that would help expand the individual parts, I recommend the manuscript for publication.

Author Response

The manuscript is devoted to a topic that is beneficial for practice. The manuscript is processed well. I consider the only "shortcoming" to be the excessive brevity of all passages of the manuscript and the relatively low number of probands. As for other shortcomings or limits of work, the authors are fully aware of them and state them in passage 4.1

Author response: Thank you for pointing this out. We have expanded our manuscript in several places, based on the comments from each reviewer. As a result, we feel this has greatly improved the manuscript quality and transparency.

My only recommendation is - to expand the introduction with more emphasis on the state of the topic, the work would benefit from the inclusion of higher number of sources.

Author response: Thank you for this recommendation. We have expanded the introduction and included more sources.

Lines 35-39

One study reported that adolescent athletes returning to sport after their concussion had a 34% increased risk of subsequent musculoskeletal injury than those who did not sustain a concussion, even after they have received return to play clearance (i.e. permission to return to full contact game play or participation) from a medical provider[7].

Lines 50-54

Current recommendations provided by the most recent consensus statement on concus-sion in sport suggest that after a brief (24-48 hour) period of strict rest post-injury, clini-cians should recommend gradually adding more activity that does not exacerbate symptoms in a graduated and stepwise fashion [2].

Lines 223-229

Interestingly, the values obtained for those who took ≥28 days to receive RTP clearance are similar to a past study, where researchers observed an average count of approximately 6,600 steps/day in the days following a concussion [29]. Furthermore, a systematic review reported normative values among boys and girls to range on average 12,000 to 16,000 steps/day and 10,000 to 13,000 steps/day respectively [25]. Across our sample, the majority of adolescents with a concussion were typically below this normative level (67% of boys, 64% of girls).

The second recommendation concerns figures - their quality does not seem to be fully satisfactory, but it is possible that in the case of a final publication, better quality will be used.

Author response: We agree with the reviewer. Using the full figure image, the figure quality improves. However, as the template requires the figure to be embedded within the word document, there is decreased quality with the figures. We have ensured the actual figure files are submitted for maximum quality, and will work with the journal editorial office to ensure they are viewed properly by the readership.

After a precise reading of the entire manuscript and making minor adjustments that would help expand the individual parts, I recommend the manuscript for publication.

Author response: Thank you for your review and for strengthening our work.

Reviewer 5 Report

It is a clear and well-written manuscript. However, the instruments used to measure physical activity and contusion symptoms are somewhat simplistic and with a low-definition capacity to explain the modifications in the variables studied.

It is important to mention in methods how they chose the cut-off points to determine the sensitivity, specificity, and accuracy for each variable. This is because these are very low and do not distinguish the possible effect of the independent variables on the early or late recovery of the athletes. The Youden index is the one that is recommended. This low sensitivity and specificity could be due to the low power of measurement in the instruments used. The latter should also be included in limitations.

The results do not support the conclusion about the beneficial effect of exercise for speedy recovery in patients with concussion. To mention it in this way is to create unnecessary myths in the literature. As the authors mention and discuss, they are simple associations, which may or may not make sense.

Lines 101-102. Set the internal consistency and test-retest reliability values of the PCSI.

Lines 127-128. Put the validity and reliability values of the instrument.

Author Response

It is a clear and well-written manuscript. However, the instruments used to measure physical activity and contusion symptoms are somewhat simplistic and with a low-definition capacity to explain the modifications in the variables studied.

Author response: We appreciate the concern regarding the limited utility of the instruments. We sought to use an ecologically valid approach to address our research question in order to improve the clinical translation of our findings. While physical activity has emerged as an important consideration in post-concussion care, there are relatively few studies that use objective methods (i.e., actigraphy) to track post-concussion activity, as most rely on subjective, patient-reported data. Thus, although simplistic, we feel our work advances the knowledge on this topic. In addition, we selected the post-concussion symptom inventory to track post-concussion symptoms, given its use in other studies in this field as well as its inclusion as a NIH/NINDS common data element for symptom tracking.

It is important to mention in methods how they chose the cut-off points to determine the sensitivity, specificity, and accuracy for each variable. This is because these are very low and do not distinguish the possible effect of the independent variables on the early or late recovery of the athletes. The Youden index is the one that is recommended. This low sensitivity and specificity could be due to the low power of measurement in the instruments used. The latter should also be included in limitations.

Author response: The cutoff points were identified on a stepwise basis using ROC analyses. We then selected the value with the highest accuracy to report. We have clarified this in our methods section.

Lines 148-153

We then used a Receiver Operating Characteristic (ROC) and Area under the Curve (AUC) analysis to identify different cutpoint thresholds to distinguish between groups for each exercise/activity variable. Using a stepwise approach, we identified the sensitivity and specificity for the value that demonstrated the highest classification accuracy to distinguish between groups for each variable.

In addition, we agree that the low sensitivity and specificity may be due to our relatively small sample size. We have added this as a limitation.

Lines 320-322

Finally, given the relatively small sample size, the low sensitivity and specificity we observed may have been due several factors, including the low power of measurement in the instruments used.

The results do not support the conclusion about the beneficial effect of exercise for speedy recovery in patients with concussion. To mention it in this way is to create unnecessary myths in the literature. As the authors mention and discuss, they are simple associations, which may or may not make sense.

Author response: Thank you for pointing this out. We have modified the conclusion to better reflect our design and results.

Lines 324-328

Among adolescents with concussion, we observed an association between returning to unrestricted sports participation within 28 days of injury with more steps/day, exercise sessions/week, and spent more total time exercising/week. These preliminary results from our cohort study further support the potential benefits of physical activity during concussion recovery, although our study is designed to evaluate association, not causation.

Lines 101-102. Set the internal consistency and test-retest reliability values of the PCSI.

Author response: We have included the internal consistency and test-retest reliability values of the PCSI in our revised methods.

Lines 101-103

The PSCI has been developed and validated specifically within the youth and adolescent population [16,17], and contains strong internal consistency (α=0.94) and test-retest reliability (ICC=0.79).

Lines 127-128. Put the validity and reliability values of the instrument.

Author response: We have added the reported validity and reliability values of this instrument in the revised methods.

Lines 134-136

A systematic review on wearable devices determined that this measurement approach demonstrated acceptable validity (±10% measurement error) and reliability (correlation coefficient = 0.75) [22].

Round 2

Reviewer 3 Report

Thanks to the authors for their careful and  thorough respones to the reviewers' suggestions

Reviewer 5 Report

None